# A Deep Learning-Based Platform for Workers’ Stress Detection Using Minimally Intrusive Multisensory Devices

**DOI:** 10.3390/s24030947

**Published:** 2024-02-01

**Authors:** Gabriele Rescio, Andrea Manni, Marianna Ciccarelli, Alessandra Papetti, Andrea Caroppo, Alessandro Leone

**Affiliations:** 1National Research Council of Italy, Institute for Microelectronics and Microsystems, 73100 Lecce, Italy; andrea.caroppo@cnr.it (A.C.); alessandro.leone@cnr.it (A.L.); 2Department of Industrial Engineering and Mathematical Sciences, Marche Polytechnic University, 60131 Ancona, Italy; m.ciccarelli@staff.univpm.it (M.C.); a.papetti@staff.univpm.it (A.P.)

**Keywords:** stress detection, smart systems, deep learning, workers’ health, sensors

## Abstract

The advent of Industry 4.0 necessitates substantial interaction between humans and machines, presenting new challenges when it comes to evaluating the stress levels of workers who operate in increasingly intricate work environments. Undoubtedly, work-related stress exerts a significant influence on individuals’ overall stress levels, leading to enduring health issues and adverse impacts on their quality of life. Although psychological questionnaires have traditionally been employed to assess stress, they lack the capability to monitor stress levels in real-time or on an ongoing basis, thus making it arduous to identify the causes and demanding aspects of work. To surmount this limitation, an effective solution lies in the analysis of physiological signals that can be continuously measured through wearable or ambient sensors. Previous studies in this field have mainly focused on stress assessment through intrusive wearable systems susceptible to noise and artifacts that degrade performance. One of our recently published papers presented a wearable and ambient hardware-software platform that is minimally intrusive, able to detect human stress without hindering normal work activities, and slightly susceptible to artifacts due to movements. A limitation of this system is its not very high performance in terms of the accuracy of detecting multiple stress levels; therefore, in this work, the focus was on improving the software performance of the platform, using a deep learning approach. To this purpose, three neural networks were implemented, and the best performance was achieved by the 1D-convolutional neural network with an accuracy of 95.38% for the identification of two levels of stress, which is a significant improvement over those obtained previously.

## 1. Introduction

Industry 4.0 implies a substantial change in production processes and worker roles, demanding the integration of human operators into new production paradigms in a socially sustainable manner [1]. Automation is relieving workers of physical efforts, but it introduces mentally demanding tasks as part of their increased responsibilities. This transition is particularly challenging for older workers who must adapt to the new environment, resulting in increased stressors and mental health risks [2]. Work-related stress significantly contributes to individuals’ overall stress levels, resulting in long-term health problems and detrimental effects on quality of life, corporate entities, and national economies. Even if psychological questionnaires have been commonly used to assess stress, they lack real-time or continuous monitoring capabilities, making it difficult to identify causes and challenging work activities. Moreover, a discrepancy often exists between self-reported stress and measured stress [3]. To address this, analyzing physiological signals such as skin temperature, breathing rate, blink detection, and the human voice can provide an efficient solution. A systematic review of the literature shows how a multimodal approach is preferable [4]. These physiological signals can be continuously measured using wearable or ambient sensors. While ambient sensor-based monitoring technologies are less obtrusive, they require a complex environmental design for reliable data. In recent years, there has been a rise in minimally intrusive wearable sensors, allowing for the monitoring of health indicators and the proactive prevention of hazardous incidents [5,6]. However, challenges related to the stability and accuracy of acquired signals must be addressed for effective use in real-world settings [7].

Wearable devices for monitoring vital signs have been developed, with cardiac and electrodermal activity analysis showing promise in assessing physical and cognitive stress [8,9]. Cardiac activity (CA) can be assessed using electrocardiography (ECG) or plethysmography (PPG). ECG provides accurate measurements but is limited to the chest area, making it suitable only for chest straps or patches. In contrast, PPG—although less accurate due to motion artifacts—is less intrusive and versatile as it can be applied to different body points and requires only a single point of contact with the skin. Electrodermal activity (EDA) measurements assess the electrical characteristics of the skin between two points using various types of electrodes. Textile and metal electrodes are more comfortable but are prone to noise and artifacts, while pre-gelled electrodes offer better stability and lower impedance.

To effectively monitor stress, wearable devices should be comfortable, non-interfering, allow real-time data access, and provide accurate measurements. Several devices meeting these criteria are available, but only a few can measure both CA and EDA. Monitoring points commonly used for EDA and CA assessment are fingers, chest, and wrist. However, these locations may not be ideal for long-term stress assessment in the work environment due to movement and intrusiveness. The wrist is subject to continuous movement, finger monitoring is intrusive, and the chest may not be optimal for EDA signal assessment.

To overcome these limitations, in our previous work, a sensorized garment was developed to concurrently monitor CA and EDA signals [10]. The sensors were strategically positioned on the body to obtain a good performance in terms of signals acquisition, minimizing the impact of movements and ensuring uninterrupted work activity. Leveraging these insights, the shoulder and earlobe were identified as the optimal measurement points for EDA and CA, respectively. The goal was to create a minimally intrusive wearable system that fulfills user comfort, real-time data access, and high accuracy requirements for reliable stress detection.

In addition, an ambient sensor was added to make the acquisition platform more stable and effective. Indeed, camera-based systems have shown effectiveness in terms of detecting stress and can be readily available in the workplace. They offer a low-cost solution without requiring daily user intervention. These “non-wearable sensors” measure specific parameters or features for stress evaluation from a distance without physical contact. They can be classified into physical and vision-based measures. Physical measurements involve capturing observable parameters of the human body, such as eye activity (including pupil dilation), human speech, and body postures [11,12,13]; whereas vision-based measures use imaging techniques to assess the stress level of an individual, and they can be further classified into thermal infrared (IR) imaging and computer vision-based techniques [14,15].

In terms of software, many stress detection platforms in the literature use machine learning (ML) techniques for assessing psychophysical stress, monitoring, above all, the wrist, chest and fingers. These studies mainly employ several supervised classifiers and monitor multiple parameters. Supervised ML schemes, particularly the K-nearest neighbor classifier, have shown the best performance with an accuracy of approximately 96% [16,17,18,19,20,21,22,23,24]. Although these approaches require a complex training phase and have the limitation of using labeled datasets of simulated events [25], they still exhibit a higher performance than the unsupervised systems described in the literature, which have an accuracy of less than 80% [26,27,28]. Additionally, most papers evaluate stress and no-stress conditions, with only a few considering multiple stress levels. By developing methods to detect multiple stress levels objectively, researchers and healthcare professionals can gain a more accurate understanding of an individual’s stress response. This can enable earlier intervention and personalized treatment plans.

In our previous work [10], from a software perspective, both supervised and unsupervised approaches were used, with accuracy performances of 94.9% for the former and up to 77.4% for the latter. For detecting multiple levels of stress, the best results were achieved through the supervised approach, with accuracy values of approximately 91%. Given its relevance, the main objective of this work was to improve this value. To achieve this, the following three deep learning (DL) algorithms were used and tested: one-dimensional convolutional neural network (1D-CNN), long short-term memory (LSTM), and gated recurrent unit (GRU). To perform the comparison between the algorithmic methods, the testing was conducted using the same dataset created in [10].

## 2. Data Acquisition

In this experimental study, 20 volunteer participants (9 males and 11 females), with an age range of 24 to 38 (mean age = 29.1 years), were recruited from the university. The stress induction procedure was explained in our previous work [10], in which the developed and tested multi-sensor platform was described. It is composed of a wearable and an ambient system. The wearable one is designed to detect heartbeat and EDA in the least intrusive way possible and consists of a shoulder strap equipped with an electronic device. Although wearable devices are the most commonly used method with which to collect physiological data, some studies have highlighted significant measurement errors attributed to the poor placement of the device or rapid movements by the user [29]. To mitigate physical discomfort and enhance the user experience, optimal points on the body (shoulder and earlobe) have been identified for physiological parameter acquisition. The key factors considered include suitable locations for reliable and clear signals, no restriction of the user’s movements or impediment of work activities, no discomfort during prolonged use, less susceptibility to motion artifacts, stability during data collection, and ease of sensor placement. It is important to note that the emphasis was on sensor placement rather than on belts, the wearability of which can be significantly improved. In addition, to make the acquisition platform more robust and reliable, a low-cost and readily available vision device (Camera RGB) was added for the assessment of eye blinks. Figure 1 shows a picture of the acquisition setup implemented for the data collection phase.

Prior to the test, the room setup was prepared, personal data were collected, and participants were informed about the study. They were asked to read and sign a consent form, turn off their phones, and wear the smart device.

The study involved four tasks aimed at inducing stress, with physiological data recorded throughout the test. Rest periods of 2 min were included between tasks, during which participants were exposed to relaxing stimuli such as classical music and slow-scrolling panoramic images [30]. The experimental procedure is described in Table 1.

Task 1, the Trier Social Stress Test (TSST) [31], involves a five-minute job interview presentation followed by a mental arithmetic task, where the participant is asked to count backward from 3895 in steps of 13. Task 2, the Stroop Color-Word Test (SCWT) [32], is a well-established stress induction test that measures the interference between color and word information, demonstrating the difficulty of naming colors when they conflict with word meanings. Specifically, the participant is asked to say the number associated with the color of the ink with which the color words are written. Task 3, the Math test, based on the Montreal Imaging Stress Task (MIST) [33], involves solving arithmetic challenges with a countdown timer, which increases the induced stress. Task 4, the Memory test, requires memorizing a 6 × 6 matrix of numbers and recreating the sequence with minimum tries. Judges monitored the participants’ performance, correcting mistakes by pressing a buzzer. At the end of the test, participants rated their perceived stress levels in each task on a scale of 0 (no stress) to 5 (maximum stress).

## 3. Software Framework

The software architecture developed for stress assessment includes the main steps shown in Figure 2, implemented in the Python programming language. The used features are the same as those used in the work [10], while the main focus will be on the classification phase, since a different methodology was considered.

### 3.1. Preprocessing

The primary aim of the data preprocessing phase is to minimize background electrical interference and artifacts caused by device movements. Appropriate software techniques were employed for each sensor to address this issue. In the case of the EDA signal, filtering and smoothing methods were applied to remove noise and disturbances. Specifically, a fourth-order Butterworth filter with a cutoff frequency of 5 Hz was utilized. Additionally, the input signal was convolved with a filter kernel to generate a smoothed signal. Moreover, to reduce the impact of motion artifacts and prevent inaccurate measurements, the technique described in [34] was adopted. Moreover, the EDA signal comprises two components: phasic and tonic. The phasic component represents the rapid response of skin conductance to a stimulus and is measured over a short period, often event-related. Therefore, the phasic component was extracted using a Butterworth bandpass filter with cutoff frequencies ranging from 0.16 Hz to 2 Hz. To address noise and artifacts in the PPG signal for cardiac activity analysis, the Python HeartPy library (ver. 1.2.7) [35] was employed. This library incorporates an adaptive threshold for peak detection, which adjusts to changes in PPG waveforms. Heartbeats are identified by calculating a moving average using a window of 0.75 s on both sides of each data point. To mitigate errors arising from motion artifacts and variable PPG waveform morphology, the threshold of the sequence of peak-to-peak intervals is considered. Then, a calibration procedure was implemented to ensure accurate data management and reduce detection errors caused by psychophysical variations among different users. The procedure involved measuring and recording the vital signals of interest while the user was in a resting state. The baseline of the EDA and PPG signals was determined by averaging the data acquired for 30 s during the initial phase of each data collection trial, where no external stimuli were applied.

Finally, a preprocessing stage was devised and implemented for the ambient sensor. The initial step involved detecting the human face in the captured image. In our proposed pipeline, we employed the Mediapipe library (ver. 0.8.10) [36]. This library provides real-time estimation of 468 3D face landmarks, encompassing various facial regions. Reference landmarks representing the left and right eyes were used to identify these areas. Each eye was characterized by 16 landmarks that accurately traced its contour. From these landmarks, we extracted pairs that exhibited greater horizontal and vertical distances, allowing us to determine axes in those directions. Finally, the ratio, which signifies the width of the eye’s aperture, was computed by dividing the lengths of the two obtained axes.

### 3.2. Feature Extraction and Selection

To extract and select features, attention was paid to multiple features in the time and frequency domains. These features, commonly used in stress detection analysis, were explored [16,19,23,27,37]. From these, the most significant features were identified, their dimension is one and they were computed using a 30 s sliding window for all signals.

To enhance system performance and simplify signal processing complexity, the Lasso regression technique [38] was adopted to determine the feature vector. This technique allows the automatic selection of the most useful features, discarding the unnecessary or redundant ones. In particular, those features that have a coefficient equal to 0 will be rejected. Based on the analysis of the dataset described in Section 2, the selected features for PPG, EDA, and ambient sensors are presented in Table 2.

### 3.3. Classification

DL has been widely employed for classification tasks due to its ability to automatically learn and extract complex patterns from data. By using neural networks with multiple layers, DL models can capture hierarchical representations, enabling them to discern subtle differences and make accurate predictions. This approach has been successfully applied in various domains, including image recognition, speech analysis, sentiment analysis, and fraud detection [39].

The proposed approach involves comparing the performance of 1D-CNN, LSTM, and GRU DL architectures in a classification model for multivariate time-series data.

One-dimensional (1D)-CNN is specifically designed to handle time-series sensor data by automatically extracting features and detecting patterns in a single spatial dimension. Its distinguishing feature is its focus on learning local features within each layer, which reduces computational load and makes it suitable for low-power hardware platforms. The architecture utilizes one-dimensional convolution layers, pooling layers, dropout layers, and activation functions. Configuring the network involves selecting hyperparameters such as filter size, subsampling factor, and the number of neurons.

LSTM, a recurrent neural network architecture, incorporates cells with input, forget, and output gates to control the flow of information. This enables the network to effectively process sequential data. LSTM’s distinctive feature is its memory unit and forget gate, which allow it to capture long-term dependencies. By selectively retaining relevant information, LSTM overcomes the limitations of traditional recurrent neural networks (RNNs) in handling long-term dependencies.

GRU, another RNN architecture, addresses the challenges faced by traditional RNNs, such as the vanishing gradient problem. It simplifies the architecture by combining input and forget gates into a single update gate. GRU adaptively updates the hidden state, capturing short-term and long-term dependencies efficiently. It has been successful in various applications, demonstrating effectiveness and efficiency. Compared to other gating architectures like LSTM, GRU has a simpler design. It combines the input and forget gates into a single update gate, reducing computational complexity and the number of parameters. Additionally, GRU does not require a separate cell state, making it more memory efficient.

#### Deep Learning Architectures

The DL architectures described in the previous section were developed using TensorFlow (version 2.7.0) and Python (version 3.8.10). In Figure 3, the structure for each architecture is depicted, in particular for (a) 1D-CNN, (b) LSTM, and (c) GRU. Specifically, the 1D-CNN architecture includes the following layers:Input layer: each sample includes the values of the selected features. In addition, the input values were normalized by scaling between zero and one [40] using the following equation:
(1)$$X_{scaled}=\frac{X-min}{max-min},$$
where the minimum and maximum values relate to the *X*-value to be normalized.Two one-dimensional convolutional layers are used for analysis and feature extraction along the time axis of the inputs, employing the standard rectified linear activation function (i.e., ReLU). The first layer consists of 128 hidden layers while the second one contains 512 hidden layers.Dropout layer: aims to improve accuracy and to overcome overfitting.Max Pooling Level: its goal consists in learning the most relevant data from feature vectors.Flatten layer: the input matrix is reshaped to generate a feature vector to classify the stress from the output layer.Output layer: The outputs of this linear layer are three neurons allowing the stress level classification according to three labels: 0 (no stress), 1 (level 1 stress), and 2 (level 2 stress). Level 0 refers to a relaxed state, level 1 implies low stress, and level 2 indicates high stress. In terms of perceived stress, as reported by participants themselves, high stress is associated with scores of 3, 4, and 5.

The LSTM architecture consists of the following layers:Input layer: as for 1D-CNN.Three LSTM layers: after the input layer, these three layers are added to improve the classification accuracy of our model.Three dropout layers: after every LSTM layer, a dropout layer has been added to enhance classification accuracy and to reduce overfitting.Output layer: as for 1D-CNN.

At last, the GRU contains the subsequent layers:Input layer: like the two above architectures.GRU layer: as described for the LSTM, this layer allows to increase the classification accuracy of stress level.Two dropout layers: inserted to improve classification values and to decrease overfitting.Output layer: like the two above architectures.

The optimal parameters of each DL architecture were obtained via a random search technique [41], allowing a search space of hyperparameter values to be defined by sampling points in that domain. In Table 3, the selected parameters for each architecture are shown.

**Figure 3 sensors-24-00947-f003:**
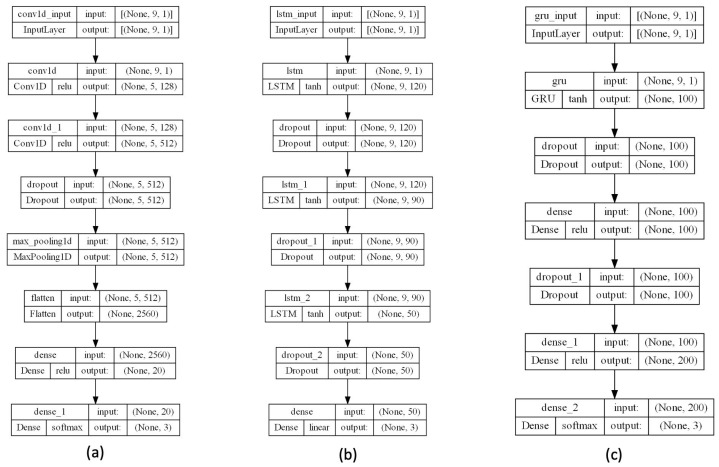
The structure for each considered DL architecture: (**a**) 1D-CNN, (**b**) LSTM, and (**c**) GRU.

## 4. Results and Discussion

To verify the goodness of the proposed approach, a series of tests were performed to verify the accuracy of the stress level classification for each considered DL architecture. The experiments were performed on an embedded PC with an Intel Core i5 processor and 8 GB RAM. Performances were evaluated according to four different metrics: accuracy (Acc), precision (Pr), recall (Re), and F1 score, defined by the following expressions:(2)$$\begin{array}{ccc} \mathrm{Acc} & = & \frac{\mathrm{TP}+\mathrm{TN}}{\mathrm{TP}+\mathrm{TN}+\mathrm{FP}+\mathrm{FN}} \end{array}$$(3)$$\begin{array}{ccc} \mathrm{Pr} & = & \frac{\mathrm{TP}}{\mathrm{TP}+\mathrm{FP}} \end{array}$$(4)Re=TPTP+FN(5)$$\begin{array}{ccc} F1-\mathrm{score} & = & \frac{2\times \mathrm{TP}}{2\times \mathrm{TN}+\mathrm{FP}+\mathrm{FN}}, \end{array}$$
where TP (True Positive) indicates the presence of a stress phase, which is correctly detected by the algorithm; FP (False Positive) indicates the absence of stress but, instead, the algorithm identifies it; TN (True Negative) indicates the absence of stress and, similarly, the algorithm does not detect it; finally, FN (False negative) indicates the presence of a stress phase but, on the contrary, it is not recognised by the algorithm, where TP (True Positive) indicates the presence of a stress phase and it is also detected by the algorithm; FP (False Positive) indicates that there is no stress phase but the algorithm detects it; TN (True Negative) means that there is no stress phase and is not detected by the algorithm; finally, FN (False negative) indicates the presence of a stress phase but the algorithm does not identify it. Accuracy shows the relation of all correctly classified samples to all samples; precision represents the model’s accuracy to provide positive occurrences; recall represents the model’s performance to find positive cases using all positive cases; F1-score influences true positive cases higher than precision.

The performance of the proposed DL architectures was evaluated on separately designed test sets. The training set was perturbed using a 10 cross-validation [42]. So, each architecture was trained using 80% of the data, while the remaining 20% were used for testing and, to avoid over-fitting, a validation set was created using 10% of the training data. The entire procedure was reiterated 10 times, training and testing with a different set in order to avoid the simultaneous appearance of the same samples in the training and test sets. Table 4 shows the results obtained for each DL architecture considering three classes (no stress, stress level 1, and stress level 2). This shows the goodness of the proposed approach, with a considerable improvement in terms of average accuracy compared to the previous work [10]. In fact, while [10] obtained an accuracy of 91% considering three classes, with the proposed approach we obtained an accuracy value varying from 92.95% with LSTM to 95.38% with 1D-CNN.

Figure 4, Figure 5 and Figure 6 depict the overall performance of the considered models by reporting model losses and accuracy during training and validation.

The goodness of the proposed approach was also assessed considering only two classes (stress/no stress). The same metrics were evaluated, and the obtained results are shown in Table 5.

Again, an improvement compared to [10] was achieved. In fact, in that case, we had obtained a maximum accuracy of 94.9% considering only two classes, whereas with the proposed approach, an accuracy ranging from 94.05% for GRU to 96.88% for 1D-CNN was achieved.

The performance improvement was mainly achieved for the multi-class system, showing that for the evaluation of multiple stress levels, the implemented hardware platform is more reliable and effective through the DL approach.

To confirm the robustness of the approach, it was tested using only the wearable sensor and only the ambient sensor with three classes. Table 6 shows the achieved results. The one-dimensional (1D)-CNN still remains the best model with an accuracy of 95.74% for the wearable sensor and of 95.16% for the ambient sensor, demonstrating that the platform also achieves a good level of accuracy when only one sensor is active, such as, for example, when the wearable sensor is unloaded.

Finally, Table 7 presents a comparison with other significant works in the literature.

Table 7 reports both wearable device-based and ambient sensor-based stress sensing systems. Information is given regarding the monitored parameters, the number of stress levels detected, the analyzed body points, and the measured accuracy values. It is evident that, despite the high accuracy values, most studies detect only one stress level. Moreover, for wearable systems, the monitored body points can be intrusive, as they often involve the fingers for EDA signals and the wrist or chest for CA. Although chest monitoring offers good accuracy and less intrusiveness, it requires the use of two devices: one to monitor EDA on the fingers and another on the chest.

## 5. Conclusions

This research paper proposes an enhanced heterogeneous multi-sensory hardware-software architecture aimed at automating the detection of stress conditions, based on our previous research. This platform holds the potential to effectively identify two levels of stress, facilitating early intervention and personalized treatment plans to improve the quality of life for workers. Two types of sensors were employed: ambient sensors and wearable devices. This approach enables versatile and efficient monitoring, adaptable to various application contexts and ensuring reliable operation even if one sensor becomes inactive or malfunctions. A user interface that could inform via an alert message—for example, a personnel manager—has not been implemented. In this configuration, the platform would act as a decision support system for the optimal management and well-being of workers.

From a hardware standpoint, the platform consists of a dual-sensor system. The wearable system was specifically designed to enable minimally intrusive monitoring and minimize disturbances caused by motion artifacts, while an easily accessible and cost-effective ambient sensor was employed. Cardiac activity, electrodermal activity, and RGB signals were considered for assessing psychophysical conditions.

To enhance performance compared to our previous techniques, a software framework was developed utilizing deep learning approaches with a high degree of generalization to ensure a good performance in real-world scenarios. The performance of the system was evaluated in controlled laboratory conditions, considering both one and two stress levels. Three deep learning algorithms were implemented and tested, with the 1D-convolutional neural network yielding the best results in terms of accuracy detection. Specifically, the accuracy values for one stress level and two stress levels were approximately 96.88% and 95.88%, respectively. These results outperformed our previous studies, particularly for the multi-stress level system, which demonstrated an improvement of 4.38 percentage points.

It is important to note that the developed software–hardware architecture was tested under laboratory conditions, and the stress induction protocol may not fully represent the actual work tasks performed by operators. A novel protocol, designed for implementation under controlled conditions, should involve the simulation of various work activities such as the assembly, quality control, and manual handling of loads. In the context of assembly and quality control, diverse stress levels could be induced by simulating escalating complexities of operations or by offering varying levels of support materials, ranging from detailed step-by-step instructions to none at all. In the case of manual load handling, the intensity of physical activity could be systematically varied. This approach also supports the differentiation between cognitive and physical stress. These limitations highlight the need for future developments to include performance evaluations in real work settings. In this context, the main challenge to address is the ground truth [45]. Below are the main strategies that will be adopted to tackle it: (i) gather data from multiple sources, in addition to physiological sensors, such as behavioral observations, expert annotations, and contextual information about the work environment, task demands, and interpersonal interactions—understanding the context will help with interpreting stress responses more accurately and differentiating between stressors; (ii) conduct longitudinal studies and continuous monitoring, rather than discrete observations, to capture variations in stress levels over time. This approach helps establish a more robust baseline and identifies patterns of stress response that may not be apparent in short-term observations; (iii) combine both subjective measures (self-reports, surveys) and objective measures (physiological data) to obtain a more accurate representation of stress and establish feedback loops with participants to refine and update the ground truth over time; (iv) explore new unsupervised algorithms that are calibrated and validated on diverse datasets. The model will be trained on a representative sample of the target population to avoid biases and improve generalizability.Furthermore, future research will focus on studying and implementing additional classification techniques, with a particular emphasis on exploring new unsupervised methods.

The proposed model is general, and while this favors its applicability, it also negatively influences its performance, as it can hardly achieve satisfactory levels of accuracy [46]. Some works highlight gender-related and age-related differences in association with work stress [47,48]; therefore, person-specific or semi-person-specific models will be investigated.

## Figures and Tables

**Figure 1 sensors-24-00947-f001:**
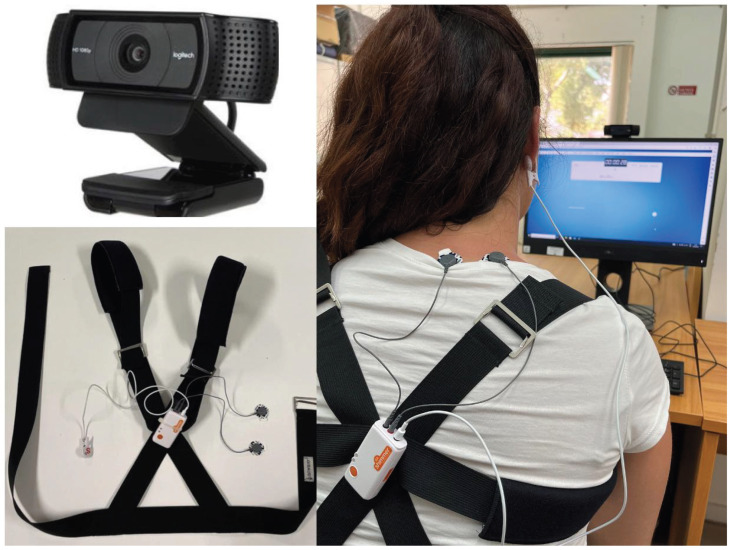
Hardware platform consisting of ambient and wearable sensors.

**Figure 2 sensors-24-00947-f002:**
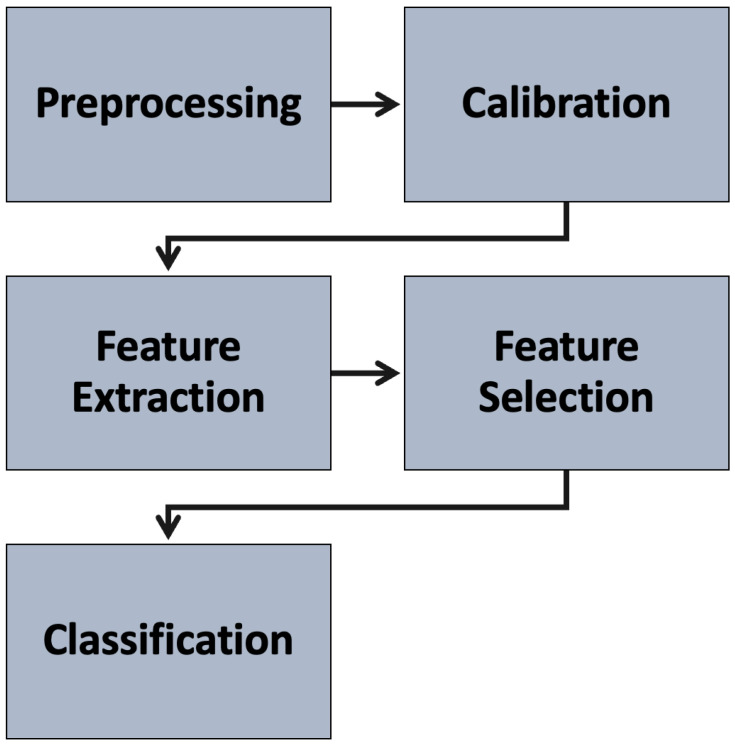
Framework software of the stress detection platform.

**Figure 4 sensors-24-00947-f004:**
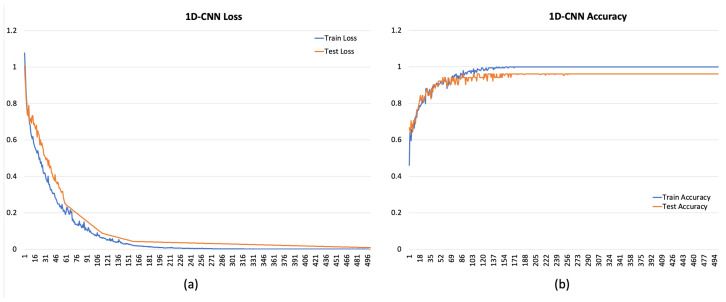
Loss (**a**) and accuracy (**b**) for 1D-CNN.

**Figure 5 sensors-24-00947-f005:**
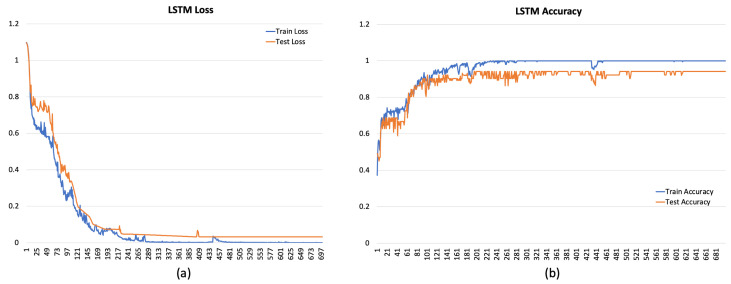
Loss (**a**) and accuracy (**b**) for LSTM.

**Figure 6 sensors-24-00947-f006:**
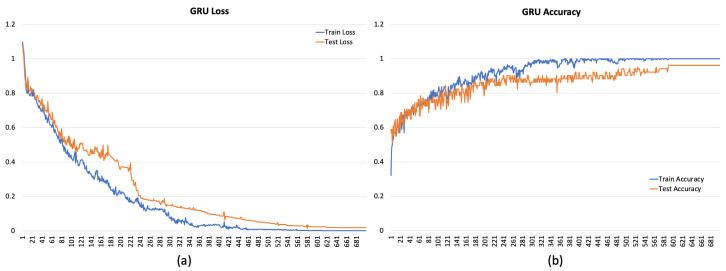
Loss (**a**) and accuracy (**b**) for GRU.

**Table 1 sensors-24-00947-t001:** Procedure of stress induction.

Task	Procedure	Time
	Baseline	2 min
1	Trier Social Stress Test	5 min
	Mental Arithmetic Stress Test	5 min
	Rest	2 min
2	Stroop Color Word Inference Test	1 min
	Rest	2 min
3	Math Test	1 min
	Rest	2 min
4	Memory Test	2 min
	Rest	2 min

**Table 2 sensors-24-00947-t002:** Features chosen for PPG, EDA, and ambient sensors.

Sensor	Feature	Description	Coefficient
PPG	RMSSD	Root Mean Square of the Successive Differences	0.867
PPG	SDNN	Standard Deviation of NN intervals	0.532
PPG	pNN50	The proportion of interval differences of successive NN intervals greater than 50 ms	0.479
EDA	GSR peak amplitude sum	GSR value at Peak-GSR value at point of onset	0.921
EDA	GSR peak energy sum	0.5 × peak amplitude × peak rise time	0.231
EDA	GSR rise rate average	Sum average of 1st derivative of points with 1st derivative >threshold (0.025)	0.416
RGB Camera	Blink number	Number of Blinks in the sliding window	0.898
RGB Camera	Ratio mean	Mean of Eye Aspect Ratio	0.654
RGB Camera	Ratio max	Maximum of Eye Aspect Ratio	0.328

**Table 3 sensors-24-00947-t003:** Selected parameters of DL architectures.

Model	Parameters
1D-CNN	optimizer = “adam”, loss_function = “sparse_categorical_crossentropy”, epochs = 500, batch_size = 64, hidden_layer_conv1d = 128, hidden_layer_conv1d_1 = 512, hidden_layer_dense = 20, dropout = 0.05
LSTM	optimizer = “adam”, loss_function = “sparse_categorical_crossentropy”, epochs = 700, batch_size = 64, hidden_layer_lstm = 120, hidden_layer_lstm_1 = 90, hidden_layer_lstm_2 = 50, dropout = 0.1, dropout_1 = 0.5, dropout_2 = 0.1
GRU	optimizer = “adam”, loss_function = “sparse_categorical_crossentropy”, epochs = 700, batch_size = 64, hidden_layer_gru = 100, hidden_layer_dense = 100, hidden_layer_dense_1 = 200 dropout = 0.05, dropout_1 = 0.05

**Table 4 sensors-24-00947-t004:** Comparison of the performance for each DL architecture.

Model	Accuracy	Precision	Recall	F1-Score
1D-CNN	0.9538	0.9603	0.9496	0.9530
LSTM	0.9295	0.9320	0.9109	0.9279
GRU	0.9505	0.9518	0.9368	0.9491

**Table 5 sensors-24-00947-t005:** Comparison of the performance for each DL architecture considering two classes.

Model	Accuracy	Precision	Recall	F1-Score
1D-CNN	0.9688	0.9708	0.9615	0.9687
LSTM	0.9499	0.9582	0.9512	0.9566
GRU	0.9405	0.9410	0.9304	0.9404

**Table 6 sensors-24-00947-t006:** Comparison of the performance for each DL architecture considering three classes and only wearable (Wear.) and ambient (Amb.) sensors.

Model	Accuracy		Precision		Recall		F1-Score	
	**Wear.**	**Amb.**	**Wear.**	**Amb.**	**Wear.**	**Amb.**	**Wear.**	**Amb.**
1D-CNN	0.9574	0.9516	0.9598	0.9589	0.9486	0.9491	0.9526	0.9531
LSTM	0.9281	0.9273	0.9361	0.9319	0.9107	0.9105	0.9189	0.9233
GRU	0.9512	0.9498	0.9507	0.9492	0.9374	0.9359	0.9487	0.9479

**Table 7 sensors-24-00947-t007:** Wearable and ambient sensor-based systems for stress detection.

Work	Vital Signs	Sensor Placement	Stress Level	Accuracy
[17]	ECG, GSR	Hand, fingers, Chest	1	0.9275
[18]	EDA, ECG, SKT	Hand, fingers, Chest	1	0.9586
[19]	ECG, EDA	Hand, fingers, Chest	1	0.875
[20]	HR, EDA, SKT	Hand	3	0.9619
[21]	PPG	Wrist	2	0.8235
[23]	PPG	Hand, fingers	4	0.9433
[24]	HR, SpO2, ST	Hand, fingers, Nose	1	0.9598
[43]	ECG	Chest	1	0.884
[13]	Physical: Eye activity (pupil dilation)	RGB Camera	1	0.9168
[44]	Physical: Human Speech	RGB Camera	1	0.9206

## Data Availability

The data presented in this study are available on request from the corresponding author. The data are not publicly available due to restrictions (their containing information that could compromise the privacy of research participants).

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
