# Peer review of "A Deep Learning-Based Platform for Workers’ Stress Detection Using Minimally Intrusive Multisensory Devices"

_sensors, 2024, doi:10.3390/s24030947_

Round 1

Reviewer 1 Report

Comments and Suggestions for Authors

Some features are listed in Table 2. Could you please append information of dimensions of the features?

Comments on the Quality of English Language

The writing of the following needs improving: "where TP (True Positive) indicates the presence of a stress phase and it is also detected by 258 the algorithm; FP (False Positive) indicates that there is no stress phase but the algorithm 259 detects it; TN (True Negative) means that there is no stress phase and is not detected by 260 the algorithm; finally, FN (False negative) indicates the presence of a stress phase but 261 the algorithm does not identify it."

Author Response

We thank the reviewer for the received positive feedbacks.

  • Question: Some features are listed in Table 2. Could you please append information of dimensions of the features?

Response: We thank the reviewer for the observation. In the revised manuscript the required information was added.

  • Question: The writing of the following needs improving: "where TP (True Positive) indicates the presence of a stress phase and it is also detected by 258 the algorithm; FP (False Positive) indicates that there is no stress phase but the algorithm 259 detects it; TN (True Negative) means that there is no stress phase and is not detected by 260 the algorithm; finally, FN (False negative) indicates the presence of a stress phase but 261 the algorithm does not identify it."

Response: Done.

Reviewer 2 Report

Comments and Suggestions for Authors

There's one missing reference I think at the top of page 7, I can't give a line number as the template seems to have messed up somehow, it's the section between line 219 and 220 before equation (1).  Otherwise I was going to suggest there needed to be more about the data capture and pre-processing but I can see that this is in your previous paper reference [9] and this gives the details I was looking for.  

Author Response

We thank the reviewer for the received positive feedbacks.

  • Question: There's one missing reference I think at the top of page 7, I can't give a line number as the template seems to have messed up somehow, it's the section between line 219 and 220 before equation (1).

Response: We thank the reviewer for the observation. In the revised version of the manuscript, the missing reference was added.

Reviewer 3 Report

Comments and Suggestions for Authors

Overview:

The paper addresses the critical issue of evaluating workers' stress levels in Industry 4.0 using a minimally invasive multisensory platform. The authors propose improvements to a previously presented system by enhancing its software performance through a deep learning approach. The results show promising accuracy, especially in detecting two levels of stress. The conclusion emphasizes the potential for early intervention and personalized treatment plans for workers' well-being.

 Review Comments and Suggestions:

The paper mentions the identification of "two levels of stress." It would be beneficial to briefly define these stress levels for better clarity, especially for readers not well-versed in the field.

The paper mentions performance evaluation in controlled laboratory conditions. How well does the proposed system generalize to diverse populations and various occupational settings? Discussing potential variations in stress detection across different demographics could enhance the paper's applicability.

The paper acknowledges the limitations of laboratory conditions and stress induction protocols. It would be insightful to discuss potential challenges or adaptations when implementing the proposed system in real-world work settings. How might the system perform in dynamic, unpredictable environments?

While the paper emphasizes a "minimally invasive" approach, it would be helpful to clarify the specific aspects of invasiveness that have been minimized. Is it related to physical discomfort, interference with work tasks, or other factors? Providing a concise definition would enhance reader comprehension.

While the dual-sensor system is highlighted, the paper briefly mentions adaptability if one sensor malfunctions. Could you elaborate on how the system ensures reliable operation and accurate stress detection in the event of sensor failures, especially in practical scenarios?

The paper discusses minimizing disturbances caused by motion artifacts. Could the authors delve into how user experience was considered in the design, especially concerning the wearable system, to ensure comfort and acceptance among workers?

The paper states an improvement over previous studies, but a brief comparison with existing stress detection methods (even if not directly related to the current research) could strengthen the context. How does the proposed platform compare to other state-of-the-art stress detection systems in terms of accuracy and practicality?

The stress induction protocol is acknowledged as potentially not representing actual work tasks. Could the authors elaborate on how they intend to address this limitation in future research to ensure the external validity of their findings?

Given the reliance on deep learning, it would be beneficial to briefly discuss the interpretability of the chosen model, particularly the 1D-Convolutional Neural Network. How easily can the system's users understand and trust the model's decisions?

Review Questions:

How well does the proposed system generalize to diverse populations, considering potential variations in stress detection across different demographics and occupational contexts?

Can the authors briefly define the two levels of stress mentioned in the paper to offer readers a clearer understanding?

Considering the focus on laboratory conditions, how might the proposed platform address challenges or variations in real-world work settings?

Could the authors provide a concise definition of what aspects of invasiveness have been minimized in the proposed system, such as physical discomfort or interference with work tasks?

Could the authors elaborate on how the dual-sensor system ensures reliable operation, especially when one sensor becomes inactive or malfunctions?

In the context of minimizing disturbances, how was user experience considered in the design of the wearable system to ensure comfort and acceptance among workers?

Given the potential limitations of the stress induction protocol, how do the authors plan to address this in future research to ensure the external validity of their findings?

Considering the reliance on deep learning, particularly the 1D-Convolutional Neural Network, how easily can users interpret and trust the model's decisions?

In addition to improvements over previous studies, how does the proposed platform compare to other contemporary stress detection methods in terms of accuracy and practicality?

Author Response

We thank the reviewer for the received positive feedbacks.

  • Question: How well does the proposed system generalize to diverse populations, considering potential variations in stress detection across different demographics and occupational contexts?

Response: We thank the reviewer for the question. In the revised manuscript the required information has been added in “Conclusions” section.

  • Question: Can the authors briefly define the two levels of stress mentioned in the paper to offer readers a clearer understanding?

Response: We thank the reviewer for the question. In the revised manuscript the required information has been added in section 3.3.1 “Deep Learning Architectures”.

  • Question: Considering the focus on laboratory conditions, how might the proposed platform address challenges or variations in real-world work settings?

Response: We thank the reviewer for the interesting question. In the revised manuscript the required information was added in “Conclusions” section.

  • Question: Could the authors provide a concise definition of what aspects of invasiveness have been minimized in the proposed system, such as physical discomfort or interference with work tasks?

Response: We thank the reviewer for the question. In the revised manuscript the required information has been added in “Data acquisition” section.

  • Question: Could the authors elaborate on how the dual-sensor system ensures reliable operation, especially when one sensor becomes inactive or malfunctions?

Response: In the revised manuscript, a table (Table 6) was added in the section 4 “Results and Discussion”, showing the obtained results considering only the wearable sensor and only the ambient sensor.

  • Question: In the context of minimizing disturbances, how was user experience considered in the design of the wearable system to ensure comfort and acceptance among workers?

Response: We thank the reviewer for the question. In the revised manuscript the required information has been added in “Conclusions” section.

  • Question: Given the potential limitations of the stress induction protocol, how do the authors plan to address this in future research to ensure the external validity of their findings?

Response: We thank the reviewer for the question. In the revised manuscript the required information has been added in “Conclusions” section.

  • Question: Considering the reliance on deep learning, particularly the 1D-Convolutional Neural Network, how easily can users interpret and trust the model's decisions?

Response: The manuscript describes a multisensory platform that is able to discriminate the presence or absence of a user's stress condition (and his or her stress level). The user interface that could inform via an alert message, for example, a personnel manager, has not been implemented. In this configuration, the platform would act as a decision support system for the optimal management and well-being of workers.

  • Question: In addition to improvements over previous studies, how does the proposed platform compare to other contemporary stress detection methods in terms of accuracy and practicality?

Response: In the revised manuscript, a table (Table 7) was added at the end of section 4 “Results and Discussion”.

Round 2

Reviewer 3 Report

Comments and Suggestions for Authors

Thanks to the author for addressing the Review questions. However, kindly address the review comments and suggestions also.

Author Response

Thank you for your suggestions and comments that were considered to improve the manuscript.

Review Comments and Suggestions:

The paper mentions the identification of "two levels of stress." It would be beneficial to briefly define these stress levels for better clarity, especially for readers not well-versed in the field.

In the revision of the manuscript, this point was addressed in “Deep Learning Architectures” section (“…Level 0 refers to a relaxed state, level 1 implies low stress, and level 2 indicates high stress. In terms of perceived stress, as reported by participants themselves, low stress is associated with scores of 1 and 2; high stress is associated with scores of 3, 4, and 5”…).

The paper mentions performance evaluation in controlled laboratory conditions. How well does the proposed system generalize to diverse populations and various occupational settings? Discussing potential variations in stress detection across different demographics could enhance the paper's applicability.

In the revision of the manuscript, these aspects were addressed in "Conclusions" section. (“…The proposed model is general, and while this favors its applicability, it also negatively influences its performance, as it can hardly achieve satisfactory levels of accuracy [43]. Some works highlight gender-related and age-related differences in the association of work stress [44,45]. Therefore, person-specific or semi-person-specific models will be investigated…”).

The paper acknowledges the limitations of laboratory conditions and stress induction protocols. It would be insightful to discuss potential challenges or adaptations when implementing the proposed system in real-world work settings. How might the system perform in dynamic, unpredictable environments?

In the revision of the manuscript, this point was addressed in "Conclusions" section. (“….In this context, the main challenge to address is the ground truth [42]. Below are the main strategies that will be adopted to tackle it: (i)Gather data from multiple sources, in addition to physiological sensors, such as behavioral observations, expert annotations, and contextual information about the work environment, task demands, and interpersonal interactions. Understanding the context will help in interpreting stress responses more accurately and differentiating between stressors. (ii) Conduct longitudinal studies and continuous monitoring, rather than discrete observations, to capture variations in stress levels over time. This approach helps establish a more robust baseline and identifies patterns of stress response that may not be apparent in short-term observations. (iii) Combine both subjective measures (self-reports, surveys) and objective measures (physiological data) to obtain a more accurate representation of stress and establish feedback loops with participants to refine and update the ground truth over time. (iv) Explore new unsupervised algorithms that are calibrated and validated on diverse datasets. The model will be trained on a representative sample of the target population to avoid biases and improve generalizability…”).

While the paper emphasizes a "minimally invasive" approach, it would be helpful to clarify the specific aspects of invasiveness that have been minimized. Is it related to physical discomfort, interference with work tasks, or other factors? Providing a concise definition would enhance reader comprehension.

In the revision of the manuscript, these aspects were addressed in “Data acquisition” section. (“…The wearable one is designed to detect heartbeat and EDA in the least invasive way possible and consists of a shoulder strap equipped with an electronic device. Although wearable devices are the most commonly used method to collect physiological data, some studies have highlighted significant measurement errors attributed to the poor placement of the device or rapid movements by the user [28]. To  mitigate physical discomfort and enhance the user experience, optimal points on the body (shoulder and earlobe) have been identified for physiological parameter acquisition…”).

While the dual-sensor system is highlighted, the paper briefly mentions adaptability if one sensor malfunctions. Could you elaborate on how the system ensures reliable operation and accurate stress detection in the event of sensor failures, especially in practical scenarios?

In the revision of the manuscript, Table 6 was added in “Results and Discussion” section.

The paper discusses minimizing disturbances caused by motion artifacts. Could the authors delve into how user experience was considered in the design, especially concerning the wearable system, to ensure comfort and acceptance among workers?

In the revision of the manuscript, these matters were addressed in "Data acquisition" section. (“….The key factors considered include suitable locations for reliable and clear signals, no restriction of user’s movements or impediment of work activities, no discomfort during prolonged use, less susceptibility to motion artifacts, stability during data collection, and ease of sensor placement. It is important to note that the emphasis was on sensor placement rather than on belts, the wearability of which can be significantly improved….”).

The paper states an improvement over previous studies, but a brief comparison with existing stress detection methods (even if not directly related to the current research) could strengthen the context. How does the proposed platform compare to other state-of-the-art stress detection systems in terms of accuracy and practicality?

In the revision of the manuscript, these points were addressed in “Results and Discussion” section (Table 7).

The stress induction protocol is acknowledged as potentially not representing actual work tasks. Could the authors elaborate on how they intend to address this limitation in future research to ensure the external validity of their findings?

In the revision of the manuscript, these points were addressed in "Conclusions" section. (“…A novel protocol, designed for implementation under controlled conditions, should involve simulating various work activities such as assembly, quality control, and manual handling of loads. In the context of assembly and quality control, diverse stress levels could be induced by simulating escalating complexities of operations or by offering varying levels of support materials, ranging from detailed step-by-step instructions to none at all. In the case of manual load handling, the intensity of physical activity could be systematically varied. This approach also supports the differentiation between cognitive and physical stress. These limitations highlight the need for future developments to include performance evaluations in real work settings…”).

Given the reliance on deep learning, it would be beneficial to briefly discuss the interpretability of the chosen model, particularly the 1D-Convolutional Neural Network. How easily can the system's users understand and trust the model's decisions?

In the revision of the manuscript, this point was addressed in "Conclusions" section (“..The user interface that could inform via an alert message, for example, a personnel manager, has not been implemented. In this configuration, the platform would act as a decision support system for the optimal management and well-being of workers…”).